# Automated, stress-free, and precise measurement of songbird weight in neuroscience experiments

Yuval Bonneh[1], Avishag Tuval[1], Ido Ben-Shitrit[1], Lilia Goffer[2], Yarden Cohen[1]*

**1** Department of Brain Sciences, Weizmann Institute of Science, Rehovot, Israel, **2** Department of Physics Core Facilities, Weizmann Institute of Science, Rehovot, Israel

* yarden.j.cohen@weizmann.ac.il

## Abstract

Monitoring the health and well-being of research animals is essential for both ethical and scientific purposes. In songbirds, body weight is one of the main indicators of their overall condition, yet traditional weighing methods can be intrusive and stress-inducing, which could decrease their song rate. We developed an automated system for monitoring the weight of multiple birds in longitudinal neuroscience experiments, which often include birds tethered to data acquisition systems. Building on previous models that were mostly designed for weighing birds outdoors and in large housing cages, our design serves as a perch for the bird to stand on (a *perch-scale*) and meets the needs of neuroscience experiments by (i) minimizing cable entanglement to safely accommodate tethered birds, (ii) supporting long-term monitoring of individually-housed birds, and (iii) linking multiple devices into a unified control unit that oversees setup, calibration, and data acquisition. We deployed the system in the cages of six canaries for ten days and validated its accuracy against daily manual weighing. The precision and continuous monitoring of the perch-scale allowed observing physiological patterns such as overnight weight loss. Our system detected 22 sequences of overnight perching in five different birds, showing an average decrease of $\approx$4% of the bird's body-weight overnight. We also found that daily weight estimates, derived from perch-scale data, were within the range of daily weight fluctuations (5–10%), as they deviated by less than 5% on average when compared to the manual weights. These results validate the device's sensitivity for detecting subtle and health-related changes. By eliminating the need for manual handling of birds, this system offers a non-invasive, hands-free approach that reduces stress and improves the accuracy of health assessments. Future applications could integrate additional health metrics to provide a more comprehensive understanding of animal welfare in neurophysiology and behavioral studies.

**Data availability statement:** The datasets generated and analyzed during this study, including reproducible analysis scripts, are available at: https://github.com/NeuralSyntaxLab/perch-scale-manuscript.

**Funding:** This work was supported by a research grant from the Latin American Hub for New Scientists, by a personal research grant (N. 2401/22 to YC) from the Israel Science Foundation, and by an ERC grant (NeuralSyntax, 101170729, to YC). The funders had no role in study design, data collection and analysis, decision to publish, or preparation of the manuscript.

**Competing interests:** The authors declare no competing financial or non-financial interests.

## Introduction

Songbirds, such as canaries and zebra finches, are excellent animal models for studying the neural basis of vocal learning and motor sequence generation [1–3]. To record neural activity, researchers carry out surgical procedures, such as electrode or optical probe implantation in the brain, followed by longitudinal observations that may extend over days to months, conducted on birds that are housed in acoustic chambers and tethered to the acquisition device [4]. Such procedures require stringent protocols for housing, husbandry, and monitoring to ensure the birds' well-being while maintaining conditions conducive to singing behavior in the lab [5]. Healthy birds are more likely to produce high-quality data, and therefore this kind of protocol is not only crucial for the animals' well-being but also beneficial for the researchers, seeking reliable data that represents natural behavior.

Body weight is a widely used indicator of health and well-being in laboratory animals, offering an objective measure of physiological status [6]. Weight loss can reflect decreased appetite as a consequence of distress, fear, and pain, but can also indicate the progression of a chronic disease reflecting deterioration with increased burden for the animal [7]. Significant deviations from baseline weight, particularly a loss of 20% or more, are used as humane endpoints across species [6]. For birds, monitoring body weight can provide valuable insight into their overall condition, especially when tracked over time. Past studies have shown that adult birds tend to lose weight following changes in husbandry [4], and experience weight fluctuations during reproductive and molting cycles [8]. Birds can even show daily and seasonal variations: Daily weight fluctuations of 5 to 10% are common, typically being highest in the late afternoon and lowest after fasting at night [9]. Seasonal variations also play a role, with birds often being heavier in winter to survive harsher conditions [8]. Some birds exhibit changes corresponding to seasonal rhythms even when kept under constant laboratory conditions [4], and hormonal regulation of these processes has been demonstrated [10]. Thus, routine weight monitoring during neurophysiological studies in songbirds could yield insights into hormonal states among other factors influencing their singing behavior.

Despite the potential benefits of weight monitoring, it is not commonly reported in birdsong neuroscience studies. Manual weighing methods, such as placing birds in cloth bags for measurement [11], can induce stress and affect song rate in laboratory conditions. Therefore, in experiments where birds are housed in acoustic chambers for song recording, handling of birds is minimized to avoid altering song output. As an alternative to weighing, Yamahachi et al. showed that song rate can indicate stress levels and demonstrated that singing more than several hundred motifs per day suggests effective stress coping in zebra finches [12]. This approach proved to be very reliable in a range of stressors, such as post-surgeries, after tethering birds to the experimental setup, and during long-term isolation, but it is specific to male zebra finches and relies on their stereotyped behavior - their stereotyped song motifs and their stable daily song rate. In many other situations, song rate remains an unreliable measure of well-being. A stereotyped song motif and a stable singing rate cannot be used in other songbird species, for example in various sparrow species and in

canaries that have seasonal changes in singing rate and highly complex songs with no countable motifs. Also in zebra finches, song cannot always be used as a reliable readout: in females, who do not sing, or in males after surgical procedures targeting brain areas in the *song premotor system* that could decrease singing behavior regardless of the animal's well-being [5]. Furthermore, a song-rate measure requires a full day of recording before motifs can be counted. In contrast, weight measuring gives an instantaneous readout and can be sensitive enough to show diurnal effects. In summary, the rate of song motifs is a reliable readout of well-being in a limited set of experiments on male zebra finches. In many other cases, additional metrics are essential.

To address these challenges, we developed a *Perch-Scale* system for easy and automated weight monitoring of freely behaving songbirds, both tethered and untethered in neuroscience experiments, and validated our system's ability to detect weight changes as a marker for their well-being. Automated perch-based weighing systems have been available and in use as early as in the 1980s - including the *Small Bird Perch-Scale* for small passerines in zoos [13], field applications for large birds such as ospreys in their habitat [14], and later fully automated solutions combining perches with electronic identification for monitoring captive and wild birds [15]. These devices were primarily developed for general health monitoring, ecological studies, or husbandry, rather than the specialized needs of neuroscience experiments. Our perch-scale system is designed to meet those particular needs by: 1) minimizing the risk of cable entanglement to safely accommodate birds tethered to data acquisition devices, 2) collecting reliable, long-term weight data continuously and from multiple birds in parallel, and 3) connecting multiple devices, each monitoring the weight of one bird, to a centralized control system that handles setup and configuration. We demonstrate here that our system allows reliable tracking of individual birds' weight over weeks and that it is sensitive enough to observe the gradual decrease in body weight overnight - validating its potential to monitor the well-being of songbirds. To help test this device and its benefits beyond the specific use in our lab, we provide the complete mechanical, electronic, and software designs of our system.

## Materials and methods

### System design

**Overview.** The Perch-Scale System is a non-invasive setup designed to monitor the weight of freely-behaving birds directly while in their cage, to detect weight loss as a marker for their health condition. The weighing is conducted by placing a load-cell-based weighing device (perch-scale) inside each bird's cage. Each perch-scale device is connected to an amplifier (NAU7802, Sparkfun), which helps amplify the signal from the load cell and then sample and convert it to an I2C signal, readable with an Arduino micro-controller, from which we can extract the weight in grams. The data collection is carried out using a Python script that runs continuously on a Raspberry Pi minicomputer. This script receives data from the Arduino and stores it as '.csv' files. Each *Scale System* can connect up to eight individual perch-scales per microcontroller with the help of another breakout board (MUX), which enables communication with multiple I2C devices simultaneously. This way, multiple birds can be monitored for their weight at the same time, as each bird is assigned to an individual perch-scale placed inside its cage.

**Scale design and technical information.** The Perch-scale device (Fig 1A) is comprised of a 3D printed Perch (150mm long, 12mm dia. made out of Delrin, scratched for better grip), a 400g custom-shaped steel cylinder as a counterweight, and a load cell with a capacity of 500g (SEN14728, Sparkfun). The load cell is connected to the perch on one side and to the steel weight on the other so that the perch floats 8mm above the surface, avoiding complications with tethered birds' cables. The steel weight is carved from a 50mm dia., 16mm tall cylinder to form designated gaps for the load cell to latch to and for the load cell wires to pass under the weight (Fig 1B). For greater convenience in handling the device, the 'base' of the weight is designed to be 16mm tall and weigh $\sim$ 200g, with two additional 100g plates of the same diameter with a hole in the center screwed on top of the base, potentially achieving 300- or 400g weight to counter the weight and momentum of a bird flying on to the perch, keeping the device stable (full design is available at https://github.com/NeuralSyntaxLab/perch-scale-system/blob/main/Mechanical%20design/scale_design.md). Canaries weigh somewhere

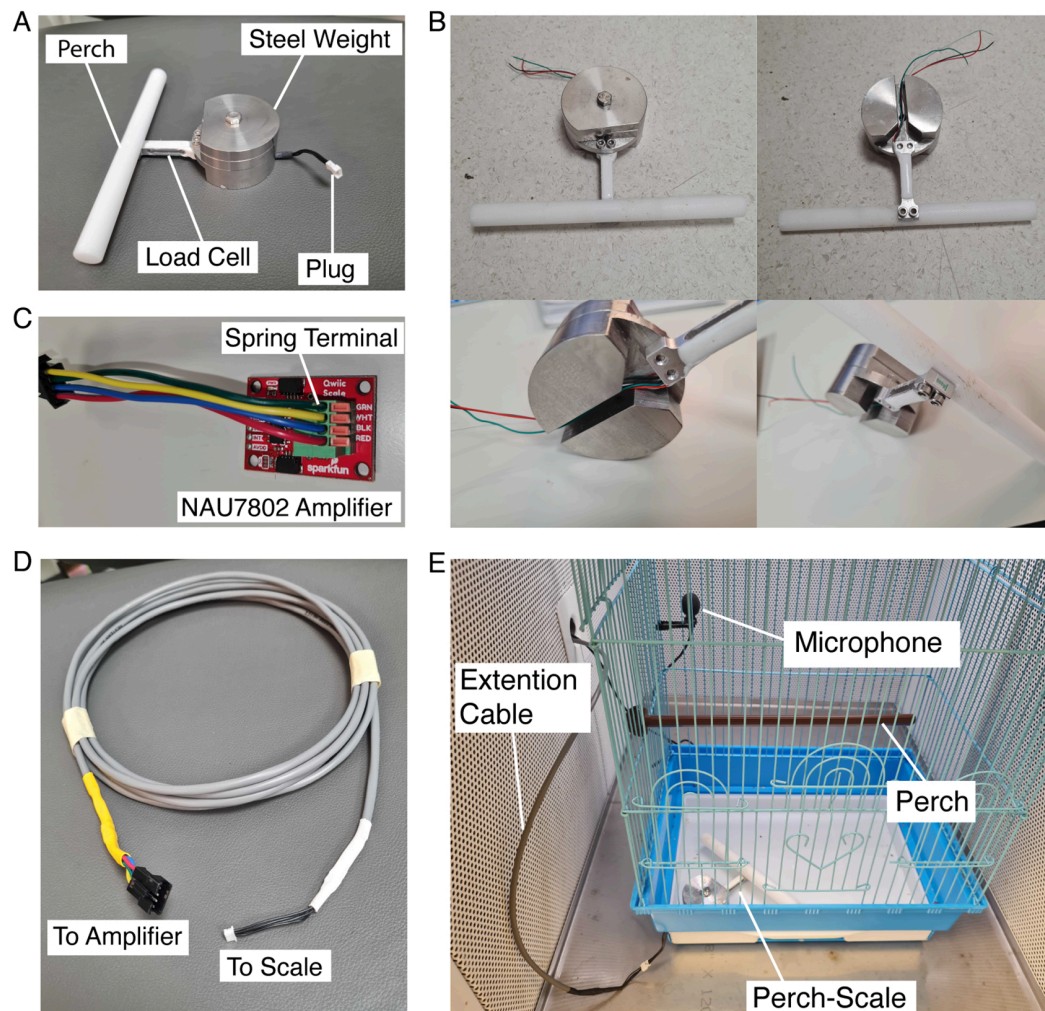

**Fig 1**. **Perch-scale system components and setup design. (A)** The perch-scale device. The perch is screwed to the load cell, which is screwed to the steel weight. The load cell's wires are soldered to the male side of the Picoblade 4-pin plug, and are passed through the gap under the weight. **(B)** Perch-scale assembly nuances depicted, including an overview, underview, and angles showing the connections between the load cell, the perch, and the steel weight. Notice the gap under the steel weight where the load cell cables go. **(C)** Image of the load cell amplifier (NAU7802 board). Notice the labeled spring terminals where the green, yellow, blue, and red wires of a JST-SM 4-PIN Pigtail connector are inserted. Here we are using extension cables to connect the load cell to the amplifier, hence the colors of the connector do not match the color labels on the board (green, white, black, and red) which are intended for the load cell's wires. **(D)** Image of the Extension Cable, soldered to the female side of the Picoblade plug, and to the male side of the JST-SM 4-PIN Pigtail connector, which will match the one that is shown in panel C inserted to the amplifier. **(E)** Image of the perch-scale setup inside an acoustic chamber, positioned at the bottom corner of the birdcage. Notice the 4-pin connector sticking out of the device in the corner of the cage, connected to the communication cables, which spread outside of the acoustic chamber through the designated hole in the side.

between 15 and 25g [16], and the perch weighs 25g, so the total weight pressing upon the load cell could reach a maximum of 50g. Regardless, we chose to use a load cell with a capacity of 500g and precision of 0.1g to withstand the force of a bird flying onto the perch with speed while maintaining accurate measures.

**Technical design.** The load cell's 4 wires (positive and negative signals – green and white, power – red, ground – black) are soldered to either plug (male or female) of a 4-pin connector (Circuit Picoblade Male-to-Female plug 425mm) as the other plug is soldered to one end of a 4-core *Extension Cable* (Fig 1D). This way, easy insertion and extraction of the perch-scale in and out of the birdcage is enabled, before connecting it to the load-cell amplifier using the long

extension cable, so that eventually the load cell's 4 wires are connected to the corresponding labeled spring terminals on the amplifier (Fig 1C). This amplifier is an Analog-to-Digital converter (ADC) with built-in gain and I2C output. This device amplifies the signal from the load cell, samples it, and converts the digital data to an I2C signal that an Arduino microcontroller can read. We can also use this signaling method to read data from multiple perch-scale devices simultaneously, with the help of another breakout board (Qwiic MUX, Sparkfun), which enables communication with up to 8 I2C addresses. Therefore, using long *Extension cables*, we can connect multiple perch-scales to one control unit (MUX, Arduino, and Raspberry Pi) and monitor the weight of up to 8 birds. This MUX breakout board is then connected to the Arduino UNO microcontroller using designated cables (Flexible Qwiic cables, Sparkfun), as the red-, black-, blue- and yellow-colored wires of these cables are connected to the 3.3V (power), Ground, SDA (serial data), and SCL (serial clock) pins in the Arduino UNO accordingly (The SDA and SCL pins are used for I2C communication). A full setup guide of the perch-scale system along with all of the components mentioned here is available at https://github.com/NeuralSyntaxLab/perch-scale-system/blob/main/User%20Guides/Scale%20System%20Setup%20Guide.md. The Arduino UNO is loaded with a specialized Arduino code using Arduino IDE. This code is designed for the Arduino to run at a baud rate of 9600 for communicating with the Raspberry Pi and transmitting weighing data from up to 8 perch-scale devices within one second. The sampling rate of the load cell amplifier is set to its maximum sample rate of 320 samples per second and each weight measurement is considered an average of 8 samples when calculating the weight in grams. These sampling rates produce an average operation time of $\sim 200ms$ for acquiring and sending precise weight data from 8 devices sequentially.

**Calibrating the perch-scales.** The load cell's output reading is a voltage signal that represents the weight. Converting this reading into accurate weight in grams, requires all perch-scale devices to be calibrated before starting to record the data. The calibration process takes place on the Arduino IDE platform, using a specialized Arduino code we developed, that uses built-in functions from the Sparkfun QwiicScale library. In the process, the system calculates the *Zero Offset* to be the number that resets the voltage reading to $0V$ when there is nothing on the load cell (except for the perch). Then, an item of known weight is placed on the perch and its weight in grams is fed to the system, which calculates the *Calibration Factor* as the division of the voltage difference from the *Zero Offset* value and the actual weight of the item. These two calibration values are stored in dedicated locations within the Arduino's Non-Volatile Memory using the EEPROM library. To allow calibrating multiple perch-scales in the same system, we use designated memory location indices to store the *Zero Offset* and *Calibration Factor* for every MUX channel. Upon connecting a new perch-scale device to a MUX channel, the user needs to re-calibrate it to store the new calibration values in place of the old ones. Whenever a new reading from a MUX channel is received, values from the channel's corresponding storage location are pulled to convert the voltage value to an actual weight value according to Eq 1.

$$\text{Weight(g)} = \frac{\text{Raw Value} - \text{Zero Offset}}{\text{Calibration Factor}} \tag{1}$$

### Experimental procedures

**Ethics declaration.** All procedures were approved by the Institutional Animal Care and Use Committees of the Weizmann Institute of Science (protocol numbers: 08891223-1, 02110223-1, and 01850223-1).

**Birds.** A total of 12 canaries (Serinus canaria, age >1 year) were monitored for their weight using the perch-scale system. Initially, as a pilot experiment for testing the perch-scale system, five birds were housed individually in cages equipped with one perch-scale device each, and placed inside an acoustic chamber (TRA Acoustics, custom-made with size $65 * 50 * 50cm$) for a few weeks, while their singing was recorded for other experiments (the *pilot* group). These birds had two other perching options in their cage as part of the standard lab protocols for housing birds. During that time, one other bird, which was used for a calcium imaging experiment, was also equipped with an individual perch-scale device in its cage to monitor its weight for nine days. This bird was tethered to a recording system while still being able to move around in its cage, without tangling the recording cable (the *tethered* bird). Lastly, six other canaries were housed

in similar conditions as in the pilot group, equipped with individual perch-scales, but had no perching options other than the perch-scale, and food and water containers hanging on the lower cage walls. These birds were monitored for their weight simultaneously for ten days, while their singing was recorded for other experiments (the *main* group). In addition to monitoring their weight with the perch-scales, daily manual weight measurements were acquired from each bird during the entire ten-day period. All birds were provided with a sufficient supply of food and water. Once a day, the doors of all acoustic chambers were opened for the birds to socialize with each other for 1–2 hours. Lighting conditions inside the acoustic chambers were similar among all birds, mimicking the outside photoperiod by turning the artificial lights on and off according to a controlled light cycle.

**Data collection.** The recording of weight data from all birds was done using a Python script that runs on a Raspberry Pi minicomputer (the control unit), which receives a list of eight data points, each referring to one weight reading from a different perch-scale, every second. This data is organized to match each of the weights to the relevant bird and then saved in .csv format with the accumulated times (*'date hh:mm:ss'* format) and weights (grams) for each bird.

During the ten-day period of simultaneous weight monitoring of the main group, we measured the weight of each of these six birds manually once a day in the afternoon (between 2-4 PM), using a cloth restraint and a standard precision laboratory balance [11] to obtain ground truth weight measurements for reference.

Each of the perch-scale devices placed in the birds' cages was calibrated once at the beginning of the experiment. One perch-scale was found to be out of tune and was re-calibrated once during the recording period (see S4 Fig panel F).

## Data analysis

**Overview.** To validate the system's reliability, we analyzed perch-scale weight data from all birds with the aim of extracting daily weight estimates comparable with the manual measurements. Since the system records continuously but only registers the bird's weight when it is on the perch, the raw dataset contains many baseline readings (near zero) when the perch is unoccupied, as well as occasional noise values due to irregular perching behavior and other factors (further shown in S1 Fig). Our analysis workflow first involved filtering out these irrelevant values, then generating two types of daily weight estimates: (1) the *daily mode*, a fast, frequency-based estimate, and (2) the *stable perch-scale estimate*, a more selective approach for identifying perched periods with minimal variation. Days (24-hour periods) that did not yield stable perch-scale estimates were excluded from the analysis, as they did not provide sufficient and reliable data for daily weight assessment.

**Filtering the data.** S1 Fig shows the distribution of raw perch-scale measurements, including the prevalence of baseline values (0–1$g$) and other ranges considered to be noise. We systematically removed all 0–1$g$ measurements when observing and presenting the results. To further avoid misleading values caused by transient noise factors, we excluded all weight measurements falling outside $\pm30\%$ of a bird's manually measured weight when calculating daily estimates. This threshold accommodates the detection of a substantial welfare red flag (defined as a drop of 20% or more from baseline weight [6]) while minimizing false deviations. An additional 10% was added to include potential measurement bias from the device. For the *daily mode* estimate, we then identified the most frequent weight measurement remaining in each day's filtered dataset.

**Stable perch-scale estimate.** To obtain a more robust estimate of the bird's weight, we also calculated the *stable perch-scale estimate*. This method uses a rolling window of 10 samples (approximately 10 seconds) applied to the filtered dataset. For each window, we computed the standard deviation (*SD*) and retained only those with $SD \leq 9\%$ of the bird's manual weight. Each qualifying window's mean value was recorded as one stable estimate. A daily weight estimate was then computed as the mean of all stable estimates within that day. Beyond producing a single daily value, this approach allows visualization of fine-scale fluctuations throughout the day, provided the bird frequently uses the perch. The choice of a 10-sample window and a 9% SD threshold was determined via grid-search hyperparameter tuning.

## Results

Individual perch-scales were installed inside the cages of 12 canaries (cf. Fig 1E). In the main group, we collected continuous perch-scale data from six of these birds for ten consecutive days, supplemented with daily manual weighing as a ground truth weight reference. These birds had access only to food and water containers as alternative perching options, placed on lower cage walls. Additionally, one tethered canary undergoing calcium imaging was monitored for nine days. We also show data from one other bird from the pilot group (see Birds), which produced valuable weight measurements for 18 days consecutively.

### Validation of perch-scale accuracy

To establish the validity of our perch-scale as a reliable weighing device, we first conducted a control experiment measuring inert objects weighing between $5 - 40\,g$ over 24 hours. The objects were continuously weighed using the perch-scale with light, humidity, and temperature conditions similar to those experienced by the canaries during the experiments. Fig 2A shows the detailed fluctuations of these measurements, demonstrating stability with most measurements within the range of approximately $0.2\,g$ around each object's true weight. From these measurements, the mode was extracted as the estimate for each object's weight and compared against their true weights, determined using a laboratory precision balance (Kern Precision Balance EWJ 300-3H; Kern & Sohn GmbH; 300 g capacity, 0.001 g readability). Fig 2B further illustrates the perch-scale's accuracy and calibration, revealing an exact linear correspondence between mode-estimated and true weights (Pearson's $r = 1.00$), along with very low measurement variability ($1SD = 0.04 - 0.08\,g$ across all tested objects). These results confirm the perch-scale's ability to provide continuous and highly accurate measurements within this weight range.

### Raw data characteristics and filtering approach

After validating the precision of the scale, we explored raw perch-scale data collected from canaries to illustrate measurement characteristics and provide a basis for our analysis approach. Fig 3A shows raw weight measurements (blue dots) from one bird across a representative daytime period. Stable weight measurements, computed here on the unfiltered

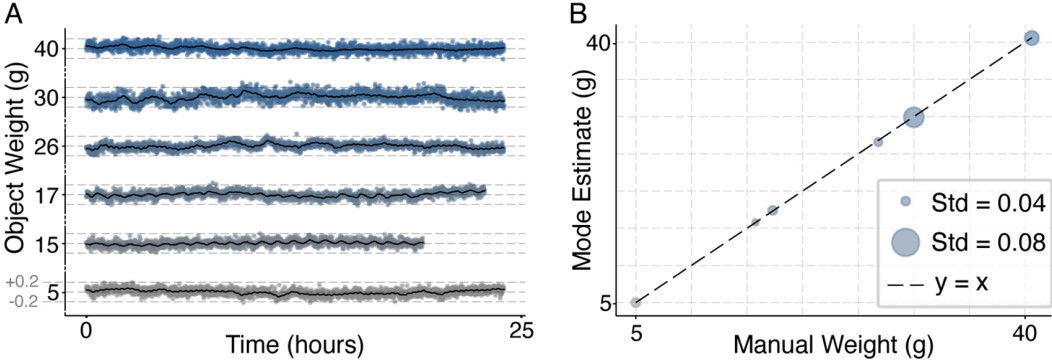

**Fig 2. Validation of the perch-scale on inert objects. (A)** Six vertically stacked time-series traces (one per object) showing $\approx 24h$ of continuous recording. Individual measurements are shown as points; a solid black line overlays each trace to show a moving-window average, which makes slow fluctuations more apparent. Three horizontal dashed guides are superimposed on each plot: the central line marks the independently measured true weight, and the upper/lower lines indicate $\pm 0.2\,g$. Across objects, readings fluctuate narrowly and symmetrically around the center. **(B)** Comparison between manually acquired true weights and perch-scale estimates (mode value of the 24-hour distribution for each item). The diagonal dashed line indicates the identity line ($y = x$). All objects fall exactly on the line (correlation = 1). Circle radius encodes the standard deviation (SD) of 24-hour recordings for each object, with 1 SD ranging between 0.04–0.08 g.

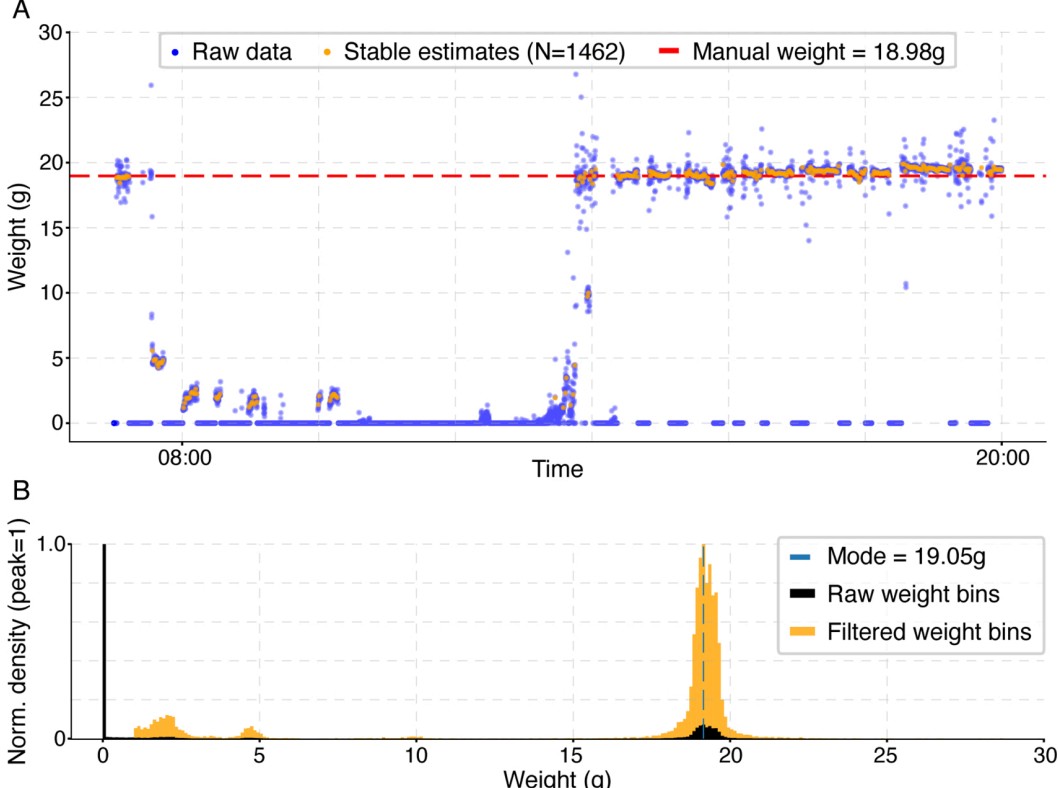

**Fig 3. Daytime example and filtering outcomes of raw weight measurements from one canary. (A)** Raw weight measurements (blue dots) collected by the perch-scale during a representative daytime period. Data points cluster around three zones: near-zero baseline readings when the bird is off the perch, a dense band near the true weight when perched, and intermediate noise values. Stable weight estimates (orange dots), derived using the filtering method but prior to applying the ±30% exclusion rule, appear mostly around the true weight but also in noisier regions. The majority of stable estimates tightly align with the manually measured ground truth weight (18.98 g; red dashed line), highlighting the effectiveness of the stable weight detection and the value of further filtering to exclude spurious detections. **(B)** Normalized histograms showing the distribution of all raw weights (black bars) and filtered weights (orange bars). Each distribution was independently normalized to its own maximum value to enable direct visual comparison. Baseline readings below 1 g were excluded from the filtered distribution, revealing a clear peak near the bird's true weight, alongside two smaller peaks that match the intermediate noise values. A vertical blue dashed line marks the mode (19.05 g), demonstrating close agreement with the ground truth weight and supporting the use of the mode for estimating daily weight.

data, are highlighted in orange dots ($N = 1462$), while the manually measured ground truth weight for the bird is indicated by a red dashed line.

The raw measurements primarily include baseline readings (0–1$g$) representing periods when the bird was not perched on the scale, clear on-scale moments around the bird's actual weight ($\approx 20g$), and some intermediate noise readings ($\approx 5g$). This distribution is further illustrated by the normalized density histogram in Fig 3B. The histogram shows raw (black bars) and filtered data distributions, with baseline readings effectively removed by applying a cutoff of <1$g$ (orange bars, see Materials and methods). After filtering, a distinct peak emerges around the bird's manually measured weight, as the vertical blue dashed line illustrates this distribution's mode estimate (19.05$g$) closely matching the manually measured weight (18.98$g$). This highlights the potential of using the mode as a reliable estimate for the bird's daily weight after applying baseline cutoffs.

Additionally, Fig 3A illustrates the effectiveness and accuracy of our stable weight measurement approach in capturing weight dynamics. After applying a ±30% filter, the remaining stable estimates ($N = 1224$) closely align with the ground truth (red dashed line) as their mode (18.99$g$) perfectly matches the ground truth weight measurement (18.98$g$).

Lower-weight readings around 0–5 g likely represent some noise (further discussed in S1 Fig), and remain excluded in this approach, reinforcing our rationale for applying a $\pm 30\%$ threshold to exclude such spurious measurements. Together, these examples showcase the properties of the perch-scale data and support our analysis heuristic for accurately capturing bird weights.

## Overnight weight loss measurements

The sensitivity and stability of the perch-scale enabled us to accurately track overnight weight loss in birds that remained perched continuously throughout the night. Fig 4A shows an example of a full overnight measurement session from one bird, including an additional hour of data before and after the session for context. This example clearly illustrates that the

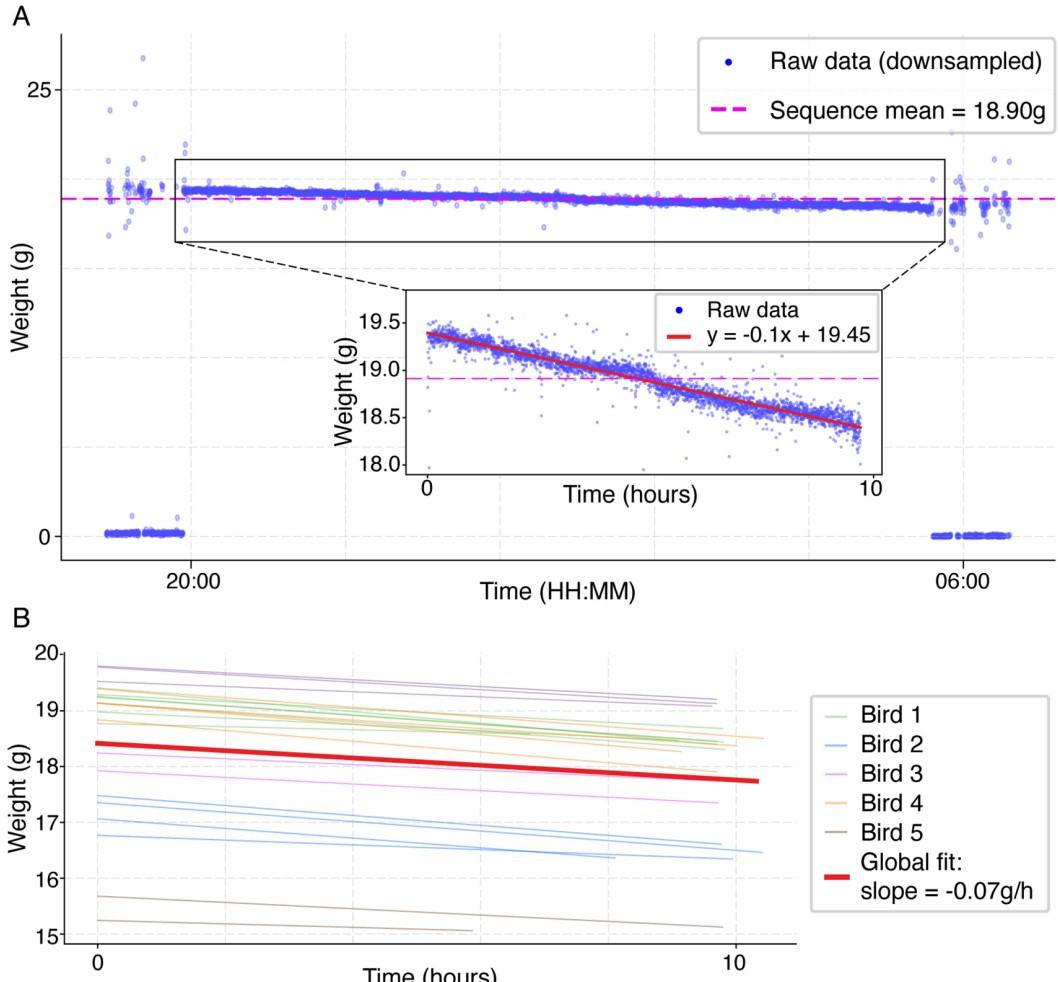

**Fig 4. Overnight weight loss trends. (A)** Example of an overnight weight change of a single bird, showing raw weight data (blue dots), including one hour of pre- and post-inactivity state for context. Once perched, the bird's weight exhibits a gradual decrease throughout the night. The magenta dashed line indicates the mean weight during the perched period (18.90 g). A zoomed-in view within the panel inset emphasizes this downward trend, with a linear regression fit (red line, $y = -0.1x + 19.45$) revealing a weight loss rate of 0.1 g/h, equivalent to 0.6% of body weight per hour and a total overnight loss of 1.15 g ($\approx 6\%$ of body weight). **(B)** Linear regression fits from 22 out of 24 overnight sessions across five different birds, with one line representing one session, as each bird's sessions are colored differently. The global fit (bold red line) shows an average weight loss rate of 0.07 g/h across all sessions. The two sessions that did not exhibit weight loss were excluded from the analysis.

weight measurements reliably shift from baseline (near zero) when the bird perches on the scale and remain consistent overnight. The inset in Fig 4A presents a closer examination of this same session, showing a gradual decrease in the bird's weight throughout the night. Linear regression analysis indicates a loss rate of approximately $0.1\,g/h$ and an overall overnight loss of $1.15\,g$, corresponding to $\approx 0.6\%$ of the bird's bodyweight per hour, and a total of $\approx 6\%$.

To evaluate the reproducibility of overnight weight loss across birds and nightly sessions, we analyzed 24 such overnight measurement sessions from different birds (Fig 4B). Linear regressions fitted to each session illustrate a clear overall trend, with an average overnight weight loss of $0.07g/h$ ($SD = 0.02g$) across all birds. Out of the 24 overnight sessions analyzed, only two failed to show such weight loss patterns (see S2 Fig), and these were excluded from the analysis. Furthermore, we explored the relationship between overnight weight loss rates and the birds' manually measured body weights. This analysis indicates a proportional overnight weight loss of approximately $0.2 - 0.5\%$ of body weight per hour for most birds, with no clear dependence on body weight within the range studied. These results validate the perch-scale's sensitivity to physiologically relevant weight fluctuations, highlighting its potential for detecting subtle health changes in birds.

## Longitudinal weight monitoring across days

We also evaluated the perch-scale system's performance in longitudinally tracking bird weights over multiple days. Fig 5A shows continuous perch-scale measurements from one representative bird (different bird from the one shown in Fig 3) across a ten-day period. Raw measurements (blue scatter points) reveal consistent diurnal weight fluctuations, while stable weight estimates (orange points) closely match the daily manual weight measurements (red lines). For this bird, we obtained on average $\bar{N} = 3319.5$ stable estimates per day with a typical fluctuation of $0.27g$ around the daily mean (Fig 5B). A closer look at the weight data from one of those days shows finer-scale fluctuations (Fig 5C). Applying a moving-average smoothing (window size = 100 samples, orange line) to $N = 4081$ stable weight estimates illustrates clear diurnal weight changes [9], with the bird's weight fluctuating by $\approx 1g$ ($\approx 5\%$ of body weight) during 24 hours: starting at $19.31g$ upon first perching in the evening, reaching a minimal weight of $18.48g$ by morning, and then returning to $19.41g$ the following evening. Such detail further demonstrates the sensitivity and accuracy of our stable weight estimation method.

To validate the accuracy of the perch-scale weight data against the daily manual measurements, we computed a *daily stable estimate* as the mean of all stable estimates within each day. We then computed the distances between each daily stable estimate and the corresponding manually measured weight. The mean of those distances is referred to here as the Mean Absolute Error (MAE). Across all days, daily stable estimates produced $MAE_{stable} = 0.64g$, with a strong Pearson correlation ($r_{stable} = 0.92$, $p < 0.001$), capturing day-to-day fluctuations. We repeated the same calculations for the daily mode estimates, resulting in $MAE_{mode} = 0.59g$ and Pearson correlation of $r_{mode} = 0.87$, $p < 0.001$. Thus, both estimating methods produce similar accuracy.

Across all six birds in the main group (see section Birds), we obtained on average $\bar{N} = 1859.5$ (range 6–5123) stable estimates per day, with a typical fluctuation of $0.43g$ around the daily mean, based on 56 successfully-obtained daily estimates out of 60 (the missing four include days when the perch-scale did not produce *Stable Perch-scale Estimates* within the filtered range at all - see Data analysis). When comparing daily *mode* and *stable* estimates to the manual weights, we observed $MAE_{mode} = 0.71g$ and $MAE_{stable} = 0.70g$, respectively. One bird's estimates were notably biased, possibly due to a calibration error (bird 2, see S4 Fig panel E, Discussion); removing this individual improved the accuracy to $MAE = 0.49g$ in both estimation methods. Additional plots for each of those birds, including a scatter plot with a summary of all comparisons, are provided in S4 Fig and panel A in S5 Fig, respectively.

Beyond the absolute accuracy, the perch-scale data also captures day-to-day weight fluctuations, as the observed daily changes closely match those of the manually acquired weights. To show that, we normalized each bird's daily estimates to their fractional changes by subtracting and dividing by the mean across days. We did the same for the manually

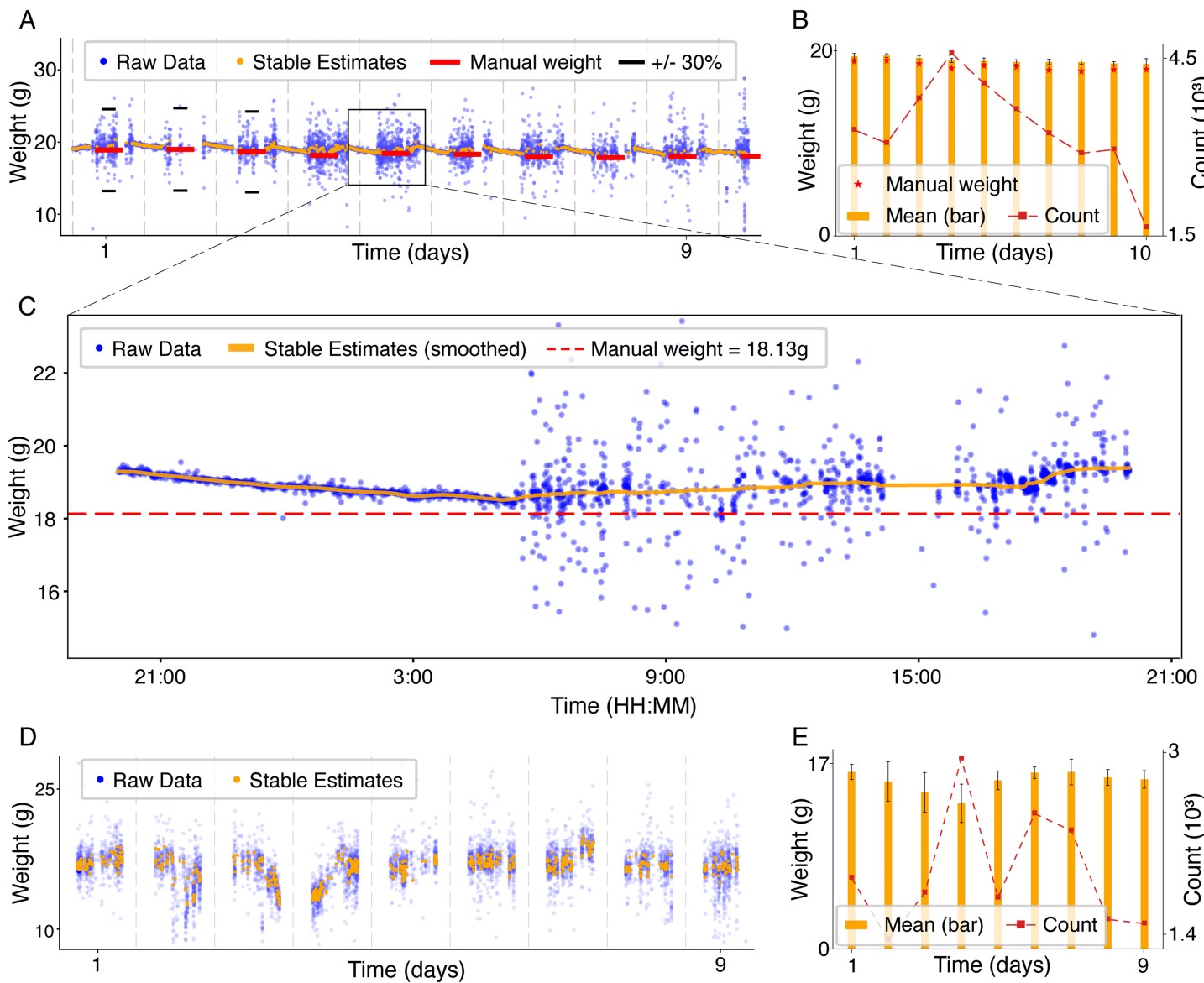

**Fig 5. Longitudinal weight monitoring across multiple days. (A)** Continuous perch-scale measurements from a representative bird over ten consecutive days. Raw weight data are shown as blue scatter points, while stable weight estimates (orange points) align closely with manual daily weight measurements (red lines). In days 1-3, black lines are added above and below the daily manual weight measurements, representing the ±30% range, which contains the majority of perch-scale data in this example. **(B)** Bar plot showing the mean of all stable estimates acquired per day for the same bird (bars, left axis) with accompanying counts of stable estimates (line, right axis). This bird produced an average of 3319.5 stable estimates per day. Error bars indicate standard deviation per day. Red stars mark the manual weight measured each day. **(C)** Expanded single-day view from the longitudinal data, showing finer-scale weight fluctuations. A moving-average smoothing of 4081 stable weight estimates (orange line, 100-sample window) illustrates clear weight changes: the bird's weight decreases from 19.31g upon first perching in the evening, reaches a minimum of 18.48g by morning, and returns to 19.41g the following evening. **(D)** Longitudinal measurements from a bird tethered to an acquisition device over nine consecutive days, showing consistent stable weight estimates and demonstrating robustness of the perch-scale system across different experimental conditions. **(E)** Bar plot showing the mean of all stable estimates acquired per day (bars, left axis) with accompanying counts of stable estimates (line, right axis) for the tethered bird shown in panel D.

measured weights, resulting in pairs of normalized estimates and normalized weights. Pearson correlation between all pairs, after removing the biased bird's data, was significant ($r = 0.42$, $p = 0.0033$, see S5 Fig panel B). These results demonstrate the robustness of the perch-scale system in acquiring accurate longitudinal weight data and support the use of both the mode and stable estimation methods as reliable approaches for daily weight assessment.

Finally, we observed similar longitudinal patterns in a tethered bird, providing reliable data for nine consecutive days (Fig 5D), as well as one more bird from the pilot group, which produced 18 consecutive days of stable weight monitoring data (S3 Fig). Taken together, these findings confirm the accuracy, reliability, and suitability of our perch-scale system for longitudinal monitoring, supporting its application in long-term health assessments within neuroscience experiments involving songbirds.

## Discussion

Monitoring the weight of animals in neuroscience experiments can help tracking their well-being and maintaining reliable physiology data collection [6]. Traditional weighing methods for songbirds, such as manually placing them in cloth bags [11], demand frequent handling and are subject to operator variability, often causing stress that disrupts natural behaviors such as singing. The design suggested here allows accurate, continuous, and reliable weight measurement without disturbing the birds' natural behavior or compromising data quality.

This weighing system builds on established perch-based weighing systems successfully used in both captive and wild contexts [13–15]. Our contributions are mainly adapting the design to meet the specific needs of neuroscience studies on tethered birds in acoustic chambers; The system delivers reliable, long-term weight monitoring, ensures the safety of birds tethered to sensitive data acquisition setups, and supports extended periods of uninterrupted, high-quality recordings of both weight and neurophysiological data. Furthermore, a single centralized control unit manages multiple devices, streamlining data collection, configuration, and calibration.

To demonstrate the Perch-Scale sensitivity, we show that the system consistently and reliably tracked overnight weight loss patterns. Birds exhibited an average weight loss rate of approximately $0.07g$ per hour ($\approx 0.4\%$ of body weight per hour, range $0.027 - 0.1g/h$). In one extreme example, a bird showed a gradual overnight decrease of 1.15g over nearly ten hours of inactivity, corresponding to $\approx 6\%$ of its body weight. Such overnight weight-loss trends were observed in 22 sessions across five birds, underscoring the sensitivity of the system to subtle physiological fluctuations and the biological relevance of this phenomenon, which aligns with previously reported daily weight cycles of $5 - 10\%$ in passerines [8].

Furthermore, our results demonstrate that the system captures accurate, real-time weight data for both tethered and untethered birds. The perch-scale exhibited high precision and reliability, closely matching manual weighing methods conducted in parallel. Specifically, across 46 daily *mode* and *stable* weight estimates obtained from perch-scale data (excluding biased data from one perch-scale), the average distance between the estimates and the daily manual weights was $0.5g$, equivalent to 2.7% of the birds' body-weight. These deviations are well within the $5 - 10\%$ bounds of daily weight fluctuations mentioned above. This level of accuracy also shows promise in successfully detecting dangerous weight-loss trends of $20\%$ or more of a bird's body-weight, as previously mentioned.

Continuous, precise weight monitoring provides insights into daily and longitudinal physiological patterns that would likely be missed with intermittent manual methods (Fig 5A–5C). This capability of the perch-scale system affords future studies investigating how environmental conditions, dietary changes, or hormonal fluctuations influence avian physiology over extended periods. Its non-intrusive design enhances welfare monitoring and data quality in avian neuroscience and behavioral research.

While this system shows some promise in future health monitoring of research birds, it is only an initial step toward a comprehensive framework for welfare assessment in longitudinal experiments. As demonstrated in studies using singing rate to assess stress [12], weight dynamics must be linked to validated indicators of health, stress, and overall well-being.

Bridging this gap will require substantial data collection and validation efforts. By releasing the complete specifications, design files, and software of our system, we aim to facilitate these future studies.

In addition to weight monitoring, future work, whose goal is comprehensive welfare assessment, should integrate complementary metrics, such as body condition scoring (BCS) [17], to provide a multidimensional picture of animal health. Combining multiple non-invasive indicators, as demonstrated in other species [18], can enhance welfare monitoring and improve our understanding of physiological responses during experimental protocols.

Despite the system's advantages, several technical and procedural considerations remain. While many results show stable and precise measurements even ten days after installation, occasional calibration drift and minor bias can occur (see examples in S4 Fig panels E, F). Although these shifts do not compromise the detection of clinically significant weight loss trends, periodic recalibration is necessary, particularly following cage maintenance or cleaning. Thus, we recommend frequent monitoring of each device to establish a consistent baseline accuracy. Additionally, noise and misleading "stable" measurements may arise from atypical bird behavior while engaging with the perch (see S1 Fig); It is possible that most of these noise factors could be eliminated by placing the perch hanging from the side of the cage walls, similar to the design in Vezina et al.(2001). When planning the new design to meet the needs of neurophysiological experiments, where the bird is often tethered to an acquisition device, one of our major concerns was to avoid cable entanglements. The new design addresses that concern by keeping the perch as close to the floor as possible, so that there is not enough space for the cable to get entangled underneath it. We show that by filtering the data obtained from this device, we can eliminate false weight measurements derived from noise. Thus, at the expense of applying a few simple filtering steps, the new perch-scale design enables working with tethered birds without compromising the safety of the bird or the acquisition device. Our suggested *mode* and *stable* estimating heuristics can further help mitigate these occurrences. Altogether, these steps can help prevent biased health assessments derived from perch-scale data, and lead to overall better accuracy in monitoring the weight of birds.

In our experiment, birds had no alternative perching options other than the perch-scale, except for the food and water containers, mounted on lower cage walls. This encouraged frequent use of the perch-scale and ensured high-quality data throughout the ten-day experiment, with some birds sleeping on the perch-scale overnight every night. In contrast, in the earlier pilot group (see Birds), where standard perches were left in place, approximately one-third of birds rarely used the perch-scale, preventing weight estimation. Based on these observations, we recommend minimizing alternative perching options in future studies employing this system.

In conclusion, our automated weighing system offers a reliable, non-invasive tool for continuous weight monitoring. By reducing the stress of traditional weighing methods, ensuring the safety of tethered birds and enabling high-resolution longitudinal data collection, this system opens new avenues for studying physiological and behavioral dynamics in songbirds under experimental conditions.

## Supporting information

**S1 Fig. Distribution of raw weight measurements across birds.** Stacked bar plots illustrate the distribution of perch-scale weight measurements across different weight ranges for each bird. In all cases, the majority of data points fall in the 0–1 g range, reflecting off-scale baseline moments when the bird was not perched. These baseline values dominate the raw dataset and highlight the need for filtering before extracting meaningful weight information. Beyond these off-scale readings, most valid measurements fall within the expected canary weight range (15–25$g$), consistent with true body weights. However, additional "noise" measurements are present outside this range, caused by factors such as transient signal fluctuations from wing flaps near the perch (0–5$g$), unstable or partial perching positions that yield misleading but relatively stable values (e.g., 5–15$g$), and rare impact artifacts producing transient, unrealistically high weights (>30$g$). In one device ('Bird 5'), loss of calibration resulted in a persistent bias toward excessively high values, further illustrating the need to monitor the calibration of these devices, especially within the first days upon setup. Together, these

distributions emphasize the rationale for applying cutoffs to exclude off-scale values and outliers, ensuring that analyses focus on reliable, stable measurements within the biological range of the birds.
(TIFF)

**S2 Fig. Outlier overnight sequences.** In these two occasions, the perch-scale does not indicate weight loss trends, as opposed to the overnight weight loss trends captured in the other 22 sessions. Panels **A-D** show a closer look into these two overnight sessions, where panels A and B relate to one sequence, and panels C and D relate to a second sequence, both from the same perch-scale and bird, on consecutive nights. It is apparent that the linear trend is somewhat unstable, especially in the case shown in panels A and B, unlike the 22 other sessions where the linear trend was stable (as shown in main Fig.4A). This could suggest that these two cases shown here fail to capture weight loss as a result of device malfunction rather than a real trend. These are brought up here to show a potential example of how noise or miscalibration affects the perch-scale data. Occurrences like this should be closely monitored to evoke recalibration of the perch-scale.
(TIFF)

**S3 Fig. Longitudinal stability of perch-scale weight monitoring during the pilot experiment.** Scatter plots (top) show raw measurements (blue) and stable estimates (orange) collected continuously across multiple consecutive days for a bird in the pilot experiment. The corresponding bar plots (bottom) summarize daily mean stable weights (bars, left axis) alongside the number of stable estimates contributing to each mean (red line, right axis). Although no manual weights were collected for direct comparison in this pilot, the data demonstrate that the perch-scale device can reliably generate stable daily weight estimates over extended periods of time, with hundreds of stable measurements per day. These results provide additional evidence for the long-term robustness and reliability of the system, complementing the main manuscript figures where manual weights were available for validation.
(TIFF)

**S4 Fig. Longitudinal weight monitoring across multiple days.** To continue the results shown in main Fig.5 of the manuscript, the longitudinal results of the remaining five birds monitored in the *main group* are shown here. **(A)** Similar to Fig.5A, an example of continuous perch-scale measurements from a representative bird, across ten days. Raw weight data are shown as blue scatter points, while stable weight estimates (orange points) align closely with the manual daily weight measurements (red lines). Black lines are added above and below the daily manual weight, representing the ±30% range used as a cutoff threshold when analyzing the daily estimates. **(B)** Summary of daily stable weight estimates for the same bird shows the mean of all stable estimates within each day (bars, left axis) with accompanying counts of stable estimates per day (line, right axis). **(C-E)** Summary of daily stable weight estimates from three other birds. The estimates summary for these birds, as well as the bird shown in panels A and B, is relatively consistent and closely matches the daily manual weights. Note here that the the data in panel E (bird 2) is consistently biased. This bird's data was removed from the summary analysis due to this bias, and this serves as an example of the need to monitor the calibration. **(F)** Scatter plot of raw data (blue points) from one perch-scale that lost calibration (bird 5) and was re-calibrated at the beginning of day 5. Red scatter points represent false weight measurements as a result of this miscalibration. These red points are scaled down by 50 to match the scale of the blue dots (right axis). The perch-scale data previous to the point of miscalibration (end of day 3) are scarce, although a few accurate measurements closely align with the mean of daily manually measured weight (dashed magenta line). The weight measurements following the recalibration process (day 5 and on) are relatively stable and closely align with the mean of daily manually measured weights. **(G)** Summary of stable estimates per day for bird 5.
(TIFF)

**S5 Fig. Accuracy of perch-scale daily weight estimates across birds.** Daily weight estimates derived from the perch-scale are compared with manual weights collected once per day. Each point represents data from one day, with the x-axis showing the manually measured weight and the y-axis showing the perch-scale estimate. Colors correspond to individual

birds, and open blue triangles indicate an outlier bird with biased measurements; removing this bird improved accuracy. **(A)** Mode-based estimates: perch-scale daily mode values plotted against manual weights. Accuracy metrics are shown with and without the outlier bird (values in parentheses indicate before removal). Removing the outlier improved the mean absolute error (MAE) from $0.71g$ to $0.49g$ and the Pearson correlation from $r = 0.79$ to $r = 0.90$. **(B)** Stable-estimate values normalized within each bird, enabling comparison of fractional daily changes independent of baseline weight differences. The linear regression fit (dashed line) demonstrates a significant positive association with manual weights ($r = 0.42$, $p = 0.0033$). This panel highlights that perch-scale data not only captures absolute weights but also tracks day-to-day fluctuations in body weight consistent with manual measurements, reinforcing its reliability for monitoring subtle longitudinal changes.
(TIFF)

## Acknowledgments

We thank the members of the Neural Syntax Lab for helpful discussions, assistance with system testing, and bird handling. We also thank the Weizmann Institute of Science's core facilities for supporting the development and implementation of this system.

## Author contributions

**Conceptualization:** Yuval Bonneh, Ido Ben-Shitrit, Yarden Cohen.

**Data curation:** Yuval Bonneh.

**Formal analysis:** Yuval Bonneh.

**Investigation:** Yuval Bonneh.

**Methodology:** Yuval Bonneh, Ido Ben-Shitrit, Lilia Goffer.

**Project administration:** Yarden Cohen.

**Supervision:** Yarden Cohen.

**Visualization:** Yuval Bonneh.

**Writing – original draft:** Yuval Bonneh, Avishag Tuval, Yarden Cohen.

**Writing – review & editing:** Yuval Bonneh, Yarden Cohen.

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
