## [Decision Letter · Decision Letter 0]

14 May 2025

PONE-D-25-19057Automated, stress-free, and precise measurement of songbird weight in neuroscience experimentsPLOS ONE

Dear Dr. Cohen,

Thank you for submitting your manuscript to PLOS ONE. After careful consideration, we feel that it has merit but does not fully meet PLOS ONE’s publication criteria as it currently stands. Therefore, we invite you to submit a revised version of the manuscript that addresses the points raised during the review process.

Please consider the Reviewers’ comments carefully and respond to each. Both agree that this system could be valuable but several questions remain. For example: it remains unclear how this system would be used for multiple birds in the same cage – so single housing would be required? Would multiple weigh-stations be needed or would the  apparatus have to be moved from cage to cage? How well would this system work for smaller birds if the variance is not proportional and there is some absolute noise?  Both Reviewers raise the issue that the mean is not the appropriate statistic for a skewed distribution. The observed weights (as mean, mode, or median) should be referenced to the actual weights of the birds in the sample presented – is there a consistent error? 

We look forward to receiving your revised manuscript.

Kind regards,

David S Vicario, Ph.D.

Academic Editor

PLOS ONE

Journal Requirements:

“This work was supported by a research grant from the Latin American Hub for New Scientists, by a personal research grant (N. 2401/22 to YC) from the Israel Science Foundation, and by an ERC grant (NeuralSyntax, 101170729, to YC).”

4. In the online submission form, you indicated that “The datasets generated and analyzed during this study are available from the corresponding author upon reasonable request.”

Reviewers' comments:

Reviewer's Responses to Questions

**Comments to the Author**

1. Is the manuscript technically sound, and do the data support the conclusions?

Reviewer #1: Partly

Reviewer #2: Yes

2. Has the statistical analysis been performed appropriately and rigorously?

Reviewer #1: I Don't Know

Reviewer #2: Yes

3. Have the authors made all data underlying the findings in their manuscript fully available?

Reviewer #1: Yes

Reviewer #2: Yes

4. Is the manuscript presented in an intelligible fashion and written in standard English?

Reviewer #1: Yes

Reviewer #2: Yes

5. Review Comments to the Author

Reviewer #1: The manuscript presents a new technique for monitoring the weights of freely moving birds. It shows potential for advancing research on avian metabolism. However, there are major issues that the authors should clarify.

Major concerns:

1. The authors housed five canaries in one cage and the sixth canary in another cage. How did they achieve weight measurements for each animal? Are all the data in Figures 2 and 3 from the singly housed bird? If so, the authors should clarify that they only measured the weight of one animal, rather than six. If not, the authors should explain how they identified which animal was measured in each data point.

2. The raw measurements varied in a range of 10~14 grams. Although the authors have shown that the scale was accurate for measuring objects, there is no ground truth measurement of the birds’ weights. The authors should compare their measurements with measurements of restrained birds, which is not difficult to do.

3. Figure 2C showed that the distribution of data is skewed. The authors chose to use the mean, rather than the mode, as the weight estimate but did not explain the rationale. There is a good reason to consider the mode as a better estimate: When animals stay still, the measured data are centralized in a narrow range and are more accurate, so it is unclear whether the estimate is biased.

Minor concerns:

1. The authors should explain in more detail the housing conditions in the method section. For example, what was the lighting condition in the aviary?

2. The authors measured the over-night weight changes in one bird three times but only presented one result. The authors should present the other two results to verify the consistency of the effect.

Reviewer #2: This manuscript addresses a relevant and important aspect of animal welfare and data collection in neuroscience experiments involving birds. The authors have developed an automated weighing system designed to minimize stress during weight measurements, offering substantial advantages over traditional manual methods. Overall, I find the work relevant, useful, and detailed. However, I have some minor suggestions for improvement and clarification:

1. Authors should cite and discuss relevant prior work, e.g. https://watchbird-ojs-tamu.tdl.org/watchbird/index.php/watchbird/article/view/1700. I also recommend a detailed literature search, it took me 1 minute to find this paper, there are possibly more.

2. The authors state that their system can monitor multiple birds simultaneously, but it appears that each bird must be housed in a separate cage. This important detail should be explicitly clarified. Since housing birds together would require individual bird identification, it would be beneficial if the authors discussed how their multi-bird system provides a distinct advantage compared to simply replicating a single-bird system multiple times.

3. All this threshold choosing and outlier removal might be unnecessary. What authors do seems to be more complicated than what is necessary. Couldn’t the authors measure the median which is robust to outliers instead of the mean? So, choosing that 10 g threshold would be unnecessary. Finally, authors should simply show all measurements rather than to arbitrarily cut off ‘off-scale’ values (<10 g) and outliers (>30 g). If they absolutely need that 10 g threshold, then they need to provide guidelines or a "recipe" for adopters of their method on how to choose that cutoff in practice. In summary, I recommend reducing the need for complicated threshold-based data cleaning.

4. The sentence stating that "two birds were excluded for not producing enough reliable data" is somewhat vague. The clearer statement from the discussion section seems to be that "from two of them (out of six) we were not able to gather substantial weight data”. The authors should explicitly indicate that these two birds presumably did not sit frequently or steadily enough on the perch to generate sufficient data. Ideally, authors quantify the fraction of time in which reliable measurements are possible.

5. The term ‘Reliable Weight Measurement’ appears more as a descriptive term rather than a clearly defined data characteristic. In addition to presenting examples of unprocessed measurements, authors should illustrate the criteria and procedures used to deem measurements "reliable."

6. I recommend showing measurements on inanimate objects first to determine system noise. Only then show measurements of animals at night where they move less, then during the day, when they move, and then when they were tethers. Z-scoring the values would allow making these measurements comparable.

7. The sentence "the canary stood on the scale for long enough" implies a duration, but the provided measure appears to be a count. Clarify whether "N̄" represents the average number of reliable measurements per day or another specific measure

8. The authors should reconsider the phrasing "natural weight loss," replacing it simply with "weight loss," unless "natural" is explicitly and meaningfully defined in the manuscript.

6. PLOS authors have the option to publish the peer review history of their article (what does this mean?). If published, this will include your full peer review and any attached files.

Reviewer #1: No

Reviewer #2: No

---

## [Author Response · Author response to Decision Letter 1]

21 Nov 2025

Dear editors, dear reviewers,

Please find our detailed rebuttal letter - addressing concerns about the manuscript.

Regarding the journal requirements stated in the Decision Letter:

1. We made sure our manuscript meets PLOS ONE's style requirements, including those for file naming.

2. We include links to all the code in the manuscript.

3. We need to add the suggested sentence ("The funders had no role in study design, data collection and analysis, decision to publish, or preparation of the manuscript.") to the financial disclosure. We added the financial disclosure clause to our cover letter as instructed.

4. We amended the Data Availability declaration to:

The datasets generated and analyzed during this study, including reproducible analysis

scripts, are available at:

https://github.com/NeuralSyntaxLab/perch-scale-manuscript

5. All data is already available at the repository we designated in the Data Availability declaration (and in answering point 4 above).

6. We reviewed our reference list to ensured that it is complete and correct.

---

## [Editor Report · Decision Letter 1]

14 Dec 2025

Automated, stress-free, and precise measurement of songbird weight in neuroscience experiments

PONE-D-25-19057R1

Dear Dr. Cohen,

Your paper has been substantially improved by substantial revisions and the inclusion of some new experimental data. We’re pleased to inform you that your manuscript has been judged scientifically suitable for publication and will be formally accepted for publication once it meets all outstanding technical requirements.

Kind regards,

David S Vicario, Ph.D.

Academic Editor

PLOS One
---

## [Editor Report · Acceptance letter]

PONE-D-25-19057R1

PLOS One

Dear Dr. Cohen,

I'm pleased to inform you that your manuscript has been deemed suitable for publication in PLOS One. Congratulations! Your manuscript is now being handed over to our production team.

Kind regards,

on behalf of

Dr. David S Vicario

Academic Editor

PLOS One